



# Hydrology without Dimensions

Amilcare Porporato

Princeton University, Princeton, USA

**Correspondence:** aporpora@princeton.edu

**Abstract.** By rigorously accounting for dimensional homogeneity in physical laws, the Pi theorem and the related self-similarity hypotheses allow us to achieve a dimensionless reformulation of scientific hypotheses in a lower dimensional context. This paper presents applications of these concepts to the partitioning of water and soil on terrestrial landscapes, for which the process complexity and lack of first principle formulation make dimensional analysis an excellent tool to formulate theories that are amenable to empirical testing and analytical developments. The resulting scaling laws help reveal the dominant environmental controls for these partitionings. In particular, we discuss how the dryness index and the storage index affect the long term rainfall partitioning, the key nonlinear control of the dryness index in global datasets of weathering rates, and the existence of new macroscopic relations among average variables in landscape evolution statistics. The scaling laws for the partitioning of sediments, the elevation profile, and the spectral scaling of self-similar topographies also unveil tantalizing analogies with turbulent fluctuations.

## 1 Introduction

Galileo is credited as the first scientist to have used dimensional analysis and scaling. In his 1638 'Dialogues Concerning Two New Sciences' (Galilei, 1914), he deduced that geometrically similar objects are not equally strong under the their own weight: "A small dog could probably carry on his back two or three dogs of his own size; but I believe that a horse could not carry even one of his own size". Since this discovery of scaling laws for complex biological materials, dimensional analysis has continued to fascinate many scientists, from Fourier to Maxwell, Reynolds, Rayleigh, Kolmogorov and Taylor, and contributed to numerous new results in several fields (e.g., Barenblatt, 1996; Szirtes, 2007; Bolster et al., 2011; Katul et al., 2019). These methods have been extremely useful in complex problems lacking closed form solutions due to nonlinear interactions and multi-physics as well as in design of experiments and numerical simulations and the rational interpretation of their results.

Looking at some of the most striking applications of the Pi theorem and self-similarity (e.g., the famous example of the atomic bomb of Taylor (Barenblatt, 1996) and the Kolmogorov spectrum of turbulence (Katul et al., 2019)), it is easy to be lured by the promise of a mathematical structure offering a powerful dimension reduction in the space of variables and parameters, even for vaguely formulated problems. In spite of the straightforward steps for its application, a brute force approach to





dimensional analysis rarely leads to useful results. These failures are probably at the origin of some of the backlash in the literature, hailing these methods as nothing more than fancy tricks, capable only of recasting already-known solutions. What is true is that applications of dimensional analysis cannot be done automatically, but require careful consideration of the problem at hand. The results, which follow from the initial hypotheses, have to be scrutinized using the the available data or simulations. In other words, when performing dimensional analysis one cannot avoid the necessary iterative process of

hypothesis formulation and subsequent verification, which forms the basis of any mathematical modeling (Logan, 2013). Its results, like a good dish, depend on its ingredients as much as on its recipe.

The presence of emerging scaling laws in geophysics has been widely recognized for a long time and the related power laws have been observed in rainfall and streamflow statistics, in landscape and river-network geometry, as well as in the aggregation properties of soils and aquifers (among many others see, e.g., Rodríguez-Iturbe and Rinaldo (2001), Gagnon et al. (2006),

Sposito (2008), and references therein). Notwithstanding these numerous examples, the presence of only a few applications of the Pi theorem in geophysics appears to be at odds with the early conclusion of Strahler that "dimensional analysis will become increasingly useful in empirical, quantitative studies in geomorphology by offering a systematic means of describing and comparing the form elements of the landscape" Strahler (1958).

In this paper, we revisit a series of fundamental problems in ecohydrology and scrutinize them under the common lens of

dimensional analysis with the goal of sharpening our intuition of the underlying physical processes. After a brief review of the concepts of dimensional analysis and scaling in Section 2, in Section 3 we consider three different instances of the partitioning of water and soils in natural landscapes. Because of their complexity, availability of global data, and the lack of governing equations from first principles, these phenomena provide a particularly fertile test-bed for dimensional techniques.

## 2    From Group Theory to Street-Fighting Hydrology

Dimensional analysis is a chapter of the more general group theory (Gilmore, 2012), based on the elegant formalism of generalized homogeneous functions (Barenblatt, 1996) and their underlying linear algebra (Logan, 2013). It formalizes the principle of dimensional homogeneity, which expresses the fact that physical equations should be valid independently of the system of units chosen. Taking this principle to its rigorous consequences allows us to exploit the dimensional symmetries of the problem in a way that may lead to useful results.

On the theoretical front, the Pi theorem provides a systematic recipe to find self-similar solutions of partial differential equations (PDEs); these underlie the local Lie groups under which the PDEs are invariant (Barenblatt, 1996; Bluman and Cole, 2012). On the more practical side, in data analysis and design of experiments (Barenblatt, 1996; Szirtes, 2007; Shen et al., 2014), dimensional analysis helps make the so-called 'Fermi reasoning' (i.e., privileging good reasoning to accuracy to achieve fair estimates) more methodical (Bhaskar and Nigam, 1990; Persico, 2004; Efthimiou and Llewellyn, 2007). By

offering a rigorous way to organize physical hypotheses about a problem, perform dimension reduction in terms of fundamental governing groups, and verify and refine the hypotheses with available information, dimensional analysis is an asset for 'street-fighting' mathematical guessing (Mahajan, 2010).





## 2.1 Scaling and power laws

The word scaling, in the sense used here, refers to the existence of a property that allows going from one scale to another
(upscaling or downscaling) using the same mathematical law and, therefore, it is related to the absence of a preferred scale or
unit of measure. From a mathematical viewpoint, this property is linked to the fact that the solutions of dimensional physical
problems are generalized homogeneous functions (Hankey and Stanley, 1972; Widom, 2009) and that dimension functions are
power-law monomials (Barenblatt, 1996).

The defining property of a homogeneous function of degree $n$,

$$f(\lambda x) = \lambda^n f(x), \tag{1}$$

makes it evident that, apart from a scale coefficient, $\lambda^n$, the same function is used to go from one scale ($x$) to another scale
($\lambda x$). The linkage to power laws becomes evident when setting $\lambda = 1/x$, which transforms the previous equation in

$$f(x) = x^n f(1) = ax^n, \tag{2}$$

where $a = f(1)$ is a constant. Power-law functions, $f(x) = ax^n$, satisfy Eq. (1).

As we will see, scaling laws appear naturally in the application of the Pi theorem and arenot limited to space and time
variables, but can involve any dimensional quantity. Scaling is often connected to fractal properties of the underlying processes
and also appear in the related fields of critical phenomena and anomalous scaling (Hankey and Stanley, 1972; Sornette, 2006),
renormalization group theory (Goldenfeld, 2018), and complete and incomplete self-similarity (Barenblatt, 1996). Sornette
(2006) has several examples of power laws ensuing from different mechanisms.

## 75 2.2 The powerful dimension reduction of the Pi theorem

The starting point of a dimensional analysis is the formulation of a 'physical law',

$$a = f(a_1, ..., a_n), \tag{3}$$

which relates a quantity of interest, $a$, the so-called governed quantity, to $n$ other governing quantities, $a_1, ..., a_n$. Eq. (3)
embeds formally our scientific hypothesis about the physical problem and serves as a mathematical placeholder to collect the
initial ingredients, on which the recipe of the Pi theorem then operates.

With (3) established, the next step is to take stock of the dimension (i.e., the factor by which a physical quantity changes
upon passage from the original system of units to another) of the quantities present in the physical law (3). According to
Maxwell's convention, the dimension of a variable $q$ is indicated as $[q]$. For example, the dimension of the velocity $v$ is
$[v] = LT^{-1}$ (as well known, in mechanics the dimensions of length mass and time – LMT – are used as a class of system of
units). As a result of dimensional homogeneity, dimension functions are always power-law monomials (Barenblatt, 1996); thus,
in mechanics a generic variable $q$ has dimension function $[q] = L^\alpha M^\beta T^\gamma$. The latter is a specific instance of the Bridgman's
equation (Bridgman, 1922; Panton, 2006), in which LMT are the chosen principal dimensions. It follows that the dimension





of the argument of transcendental functions is unity (i.e., they are dimensionless quantities), so that their numerical value is identical in all systems of units.

The number $k$ of fundamental quantities in the physical law (3) plays a key role in the Pi theorem. Such fundamental quantities, also called repeating variables, must be dimensionally independent (Barenblatt, 1996), that is none of their dimensions can be represented in terms of a product of powers of the dimensions of the remaining $k-1$ quantities (in turn, this is true only if the determinant formed with the exponents of the dimension functions is different from zero). Often $k$ is equal to the fewest independent dimensions required to specify the dimensions of all quantities involved in (3), but in a few cases the number of primary dimensions differs when variables are expressed in terms of different systems of dimensions (e.g., MLT, FLT, or any other combination). In any case the value of $k$ is given by the rank of the dimensional matrix, formed by listing all the exponents of the primary dimensions of each variable in (3); see e.g., Panton (2006); Logan (2013). Once it has been ascertained that the problem admits $k$ dimensionally independent quantities among the governing quantities, the $k$ repeating variables may be chosen and the physical law (3) may be formally re-arranged as

$$a = f(a_1, ..., a_k; a_{k+1}, ..., a_n), \tag{4}$$

where conventionally the semicolon separates the repeating variables from the other ones.

Enforcing dimensional homogeneity on Eq. (4) leads to the main outcome of the Pi theorem (Barenblatt, 1996), namely that the physical law, if true, can be written in the form

$$\Pi = \varphi(\Pi_{k+1}, ..., \Pi_n), \tag{5}$$

in terms of the dimensionless Pi groups

$$\begin{aligned} \Pi &= \frac{a}{a_1^{\alpha_1}, a_2^{\alpha_2}, ..., a_k^{\alpha_k}} \\ \Pi_{k+1} &= \frac{a_{k+1}}{a_1^{\alpha_{1,k+1}}, a_2^{\alpha_{2,k+1}}, ..., a_k^{\alpha_{k,k+1}}} \end{aligned} \tag{6}$$

$$...$$

$$\Pi_n = \frac{a_n}{a_1^{\alpha_{k,n-k}}, a_2^{\alpha_{k,n-k}}, ..., a_k^{\alpha_{k,n-k}}}.$$

A comparison of Eqs. (3) and (5) clearly shows the great achievement of the Pi theorem in re-expressing a general mathematical relationship between a quantity of interest and $n$ dimensional quantities as a new relationship between $n-k$ dimensionless quantities in a more manageable, lower dimensional space. The most striking applications in the literature are in fact linked to a drastic dimension reduction (in the famous problems of the pendulum and the atomic bomb (Barenblatt, 1996), $n = 2$ and $k = 2$ and $n = 3$ and $k = 3$, respectively so that $n - k = 0$ and one is left with a dimensionless function which is a constant!). In nonlinear PDEs this may allow transforming them into ordinary differential equations (ODEs), with much greater chances of finding a solution, either analytically or numerically. While this happens more frequently in one spatial dimension





(e.g., Barenblatt, 1996; Daly and Porporato, 2004b; Eggers and Fontelos, 2008), it is also possible in more than one spatial
dimension (Hills and Moffatt, 2000; Xue and Stone, 2020).

Depending on the governing variables involved in the physical law, there may be freedom in choosing the $k$ governing
repeating variables. Each admissible choice leads to dimensionless groups that are related to the ones obtained from a different
admissible set. As we will see, while in general the different ensuing representations are equivalent, some combination may be
more revealing of the underlying dynamics and afford a more parsimonious representation. In PDEs, this freedom may lead
to different phase-space representations, some of which could be more amenable to analysis (Gratton and Minotti, 1990; Daly
and Porporato, 2004b).

In summary, starting from a physically meaningful law (1), the Pi theorem not only provides a mathematically more specific
and elegant expression for it, but also helps reveal the actual physical controls of the problem, which emerge through it in the
form of the Pi numbers obtained: these, and not the single variables listed in the original law, are the actual quantities governing
the physical phenomenon (for example, the Reynolds, Mach, and Froude numbers and the many other ones, including those
that we will see later in this article).

### 2.2.1 Self-similarity

When one or more of the $\Pi$ groups attain very large or very small values, the function $\varphi$ in (5) may reach an asymptotic form
related to a self-similar regime. In the simplest form of self-similarity, called complete or of the first kind (see Barenblatt, 1996),
the function $\varphi$ reaches a constant plateau for either very small or very large values of the governing groups. As a result, the
physical problem does not change even if the values of these groups change and the 'self-similar group' can then be eliminated
from Eq. (5), allowing for further dimension reduction. In more complicated cases the self-similarity is of the second type, or
incomplete, because the function $\varphi$ still depends on the self-similar Pi group according to a power law with an exponent, which
is not directly related to any of the dimension functions of the governing quantities.
Thus, assuming for example self-similarity with respect to the group $\Pi_{k+1}$, Eq. (5) becomes

$$\Pi = \lim_{\Pi_{k+1} \to 0 \text{ or } \infty} \varphi\left(\Pi_{k+1}, \Pi_{k+2}, ..., \Pi_n\right) = \Pi_1^\beta \psi(\Pi_{k+2}, ..., \Pi_n), \tag{7}$$

where $\beta = 0$ in the case of complete self-similarity. Thus complete self-similarity allows us to further reduce the dimensionality
of the problem by as many dimensions as there are self-similar groups. This type of similarity is frequently encountered in near-
wall turbulence, where the global Reynolds number and the dimensionless distance from the wall appear as self-similar groups.
As we will see later on, complete self-similarity is also present in landscape channelization, when fluvial erosion dominates
over soil creep.

Incomplete self-similarity seems to be more rare and is often difficult to distinguish from the case of complete self similarity,
especially for experimental or numerical problems where there are transitions to different regimes (Spagnoli, 2005), or in
which the available data are not sufficiently precise (Barenblatt et al., 1997; Smits et al., 2011; Yin et al., 2019). In self-similar
solutions of PDEs, incomplete self-similarity typically results from a nonlinear eigenvalue problem and leads to power laws that
control the temporal or spatial behavior of the problem with exponents that are irrational numbers that are not directly related



to the dimension functions of any of the variables or initial and boundary conditions (e.g., Gratton and Minotti, 1990; Aronson and Graveleau, 1993; Barenblatt, 1996; Burton and Taborek, 2007; Zheng et al., 2014). It is also worth noting that, starting from different physical laws with or without a certain variable, one sometimes arrives at different final forms of self-similar

behavior. As we will see in the example of weathering in Sec. 3.2, it appears that such cases are actually related to problems of complete self-similarity, because the alternative formulation as an incomplete self-similarity problem is characterized by integer exponents, which allow simplifications among variables, which in turn lead to the complete self similarity obtained with the other choice of variables.

## 2.3 Augmented and directional dimensional analysis

It is not infrequent in the literature to come across a point of view, made explicit by Bridgman (1922), that 'there is nothing sacrosanct' about the choice of primary dimensions and that 'dimensional analysis is merely a man-made tool that may be manipulated at will'. This is indeed in line with modern physics, which links length, time, mass, and energy to the different descriptions of reality depending on the scale of observation. Once a free choice of the primary units is accepted, then the question remains of what choice will be of maximum utility (Moon and Spencer, 1949).

Compared to the aforementioned (Section 2.2) freedom of choosing repeating variables or classes of system of units (e.g., the length, force, and time, LFT, instead of the length, mass, and time, LMT), the augmented dimensional analysis refers to a more drastic freedom of 'inventing' the type and number of primary dimensions. It is related to the original observation of that 'dimensions of quantities do not always afford a test of their identity' (Lodge, 1888). Since this ultimately affects the number of dimensionless groups and the extent of dimension reduction, the practical implications may be significant.

As the reader may imagine, this line of reasoning has attracted both stern skepticism and enthusiastic support, leading to interesting debates and controversies, with defenders of rigor and objectivity on one side and advocates of a flexible approach on the other. In the writer's experience, the subject is always a source of interesting discussion, if not else because, when one writes in favor of it, the editors somehow always manage to select a reviewer who belongs to the skeptical camp.

The most emblematic case of controversy is perhaps the well known Rayleigh-Riabouchinsky controversy (e.g., Gibbings,

1982; Butterfield, 2001). Dealing with a problem of heat transfer between a body and a fluid stream, Rayleigh (1915a) solved it within the domain of thermodynamics, i.e., including temperature as primary dimension, and obtaining one governing dimensionless group. Riabouchinsky (1915) objected that adopting the more advanced point of view of statistical mechanics, according to which temperature is the mean molecular kinetic energy, one could more parsimoniously limit the primary dimensions to length, time, and energy, without involving temperature. As a result, he obtained two dimensionless groups, reaching

the paradoxical conclusion that an apparently more detailed knowledge of the problem yields a less informative result.

The resolution of this controversy (see Appendix A for more details), shows that it is not a matter of the arbitrariness of which dimensions are considered, but of the level of description intended for a problem. Whether to include the specifics of the molecular motion depends on the size of the considered object. If one, following Rayleigh, accepts a thermodynamic approach, then the details of the disorderly molecular energy are irrelevant and temperature can be treated as an independent quantity

compared to the kinetic energy of the mean motion. Formally, choosing a greater number of fundamental units (i.e., the number





of primary dimensions) is made possible by the addition of corresponding dimensional unifiers (Panton, 2006). The confusion often arises in those cases where the structure of the equation is such that the dimensional unifier can be tacitly eliminated or taken for granted.

Of similar nature, and perhaps even more subtle and controversial, is directional dimensional analysis, which is based on the

fact that in some cases distinguishing between vertical and horizontal dimensions provides more informative results (i.e., fewer dimensionless groups). Williams (1890) argued that 'owing to the dimensions of space, the unit of length is involved in different ways, according to the different relative directions in which it may be taken.' While for some authors it remains controversial (see, e.g., Barr, 1984; Gibbings, 1980; Kader and Yaglom, 1990, and references therein), others have worked to link these extensions of dimensional analysis more solidly to group theory (Moran and Marshek, 1972). Directional dimensional analysis

works, provided that the ratio of the length dimensions does not play a role in the physics of the problem (e.g., in the original equations); this happens, for example, in cases that assume incompressibility where one of the lengths may be simply related to mass flow, because the mass dimension has been canceled out of the equations. As we will see, the variable $z$ in the landscape evolution model, analyzed in Sec. 3.3, falls within this category.

Directional dimensional analysis and its generalizations have been used successfully in a variety of problems, including

applications in mechanics and atmospheric sciences (e.g., Huntley, 1967; Moran and Marshek, 1972; Kader and Yaglom, 1990; Siano, 1985; Daly and Porporato, 2004a, b; Dimitrakopoulos and DeJong, 2012; Bonetti et al., 2020; Hooshyar et al., 2020; Sun, 2020; Hooshyar et al., 2021). Notwithstanding we do not have rigorous criteria to assess whether and when the 'tricks' of augmented dimensional analysis can be applied, there are certainly several cases in which extending the set of primary dimensions is useful. Even in case of failure, the negative results can always be used to sharpen our physical laws and

improve our starting point for dimensional analysis.

## 3 Water and soil mineral partitioning in the critical zone

Some of the most important questions of terrestrial geophysics are related to the partitioning of water and soil minerals at the land surface. Figure 1 shows the three cases that we will analyze in this section: the partitioning of rainfall into evapotranspiration and percolation plus runoff, the soil partitioning of minerals either lost by chemical dissolution (weathering) or

transported away, and the related geomorphologic partitioning of soil over the landscape due to soil creep and fluvial erosion. Since their complexity prevents us from writing the governing equations and boundary conditions in detail, these problems are excellent candidates for dimensional analysis, applied in combination with available data and numerical simulations of simplified models. For simplicity, we focus on the dominant components of such partitionings and consider only long-term averages, assuming stationary conditions.

The scaling laws obtained from dimensional analysis shed light on the dominant soil, vegetation and climate controls for these hydrologic partitionings. Specifically, in the first application it will be seen how the dimensionless dryness and storage indices determine the long term rainfall partitioning. The second application will reveal the key nonlinear control of the dry-





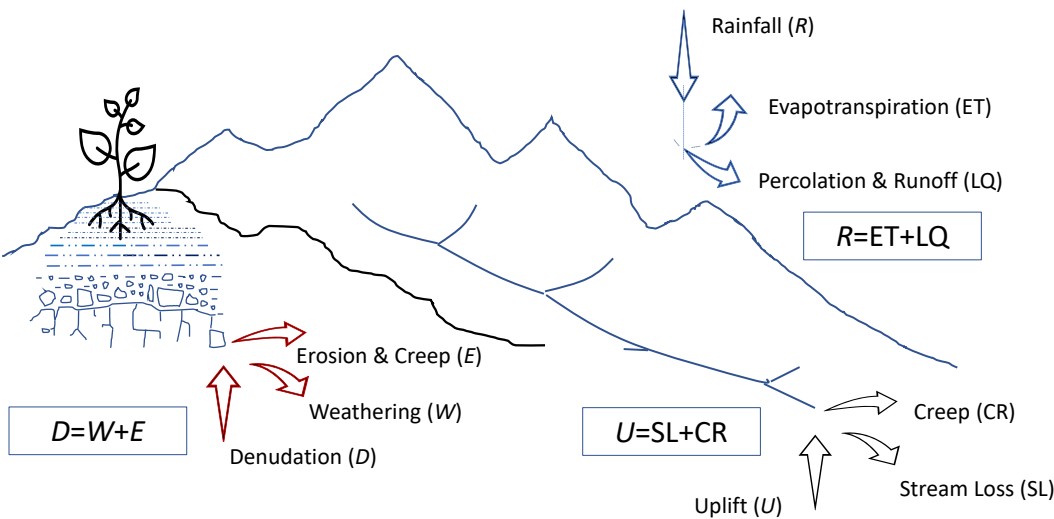

**Figure 1.** The three partitionings analyzed in this paper using dimensional analysis: the rainfall partitioning taking place at the land surface, the soil mineral partitioning by chemical dissolution (weathering) and transport processes, and the related partitioning of soil sediments responsible for landscape evolution and the formation of drainage networks.

ness index on weathering rates, while the third application will analyze new macroscopic relations among average variables landscape evolution, uncovering an intriguing analogy between self-similar topographies and turbulent flow fields.

## 3.1 Rainfall partitioning

The rainfall reaching the soil surface is either lost by runoff or infiltrates into the soil, where in turn is either lost by evapotranspiration or percolation. The fate of this partitioning is essentially controlled by the properties of the soil-plant system, which – in a sense – acts as a geophysical valve, not only for the entire hydrologic cycle, but also for the energy and carbon cycles. This fundamental hydrological problem presents robust behaviors for its macroscopic (i.e., averaged) patterns, as well as rich and complex controls at the detail level, where the role of temporal fluctuations and spatial heterogeneities becomes important.

We will indicate the mean rainfall rate as $R$ and consider together the main losses: the evapotranspiration rate (ET) and the mean rate of percolation plus runoff (LQ); see Figure 1. Assuming stationarity over long time scales and referring to a given area, the input balances the outputs,

$$R = \text{ET} + \text{LQ}. \tag{8}$$

It should be clear that, depending on the area chosen and whether the latter is homogeneous in term of land cover, soil properties, and topography, the actual meaning of the term LQ in Eq. (8) may be very different. In particular, if the control volume for our water balance is the rooting zone of a small homogeneous plot of vegetated soil, then LQ will be the average of a very intermittent term given by the sum of surface runoff and percolation to soil layers below the rooting zone, while if we consider





an entire river basin, it will be mostly the average of the streamflow draining the area. How these processes scale with the land area is an interesting open question, which is outside of our scope (see Fig. 4 by Yin et al. (2019) for an analysis of this effect). We specifically focus on ET, as many have done before, but here we adopt the special lens offered by the Pi theorem to explore the implications of different hypotheses in the physical laws used as starting points.

### 3.1.1 Turc and Budyko spaces: dryness and humidity as main controls of rainfall partitioning

While they did not use dimensional analysis, Turc and Budyko started their work by making what is perhaps the simplest hypothesis of a physical law for the rainfall partitioning (see Dooge, 1992; Daly et al., 2019, and references therein),

$$\mathrm{ET} = f_{\mathrm{TB}}(R, \mathrm{ET}_{\max}), \tag{9}$$

where $\mathrm{ET}_{\max}$ is a reference (e.g., potential or maximum) evapotranspiration. The rank of the dimension matrix

$$
\begin{array}{c|ccc}
 & \mathrm{ET} & R & \mathrm{ET}_{\max} \\
\hline
\mathrm{L} & 1 & 1 & 1 \\
\mathrm{T} & -1 & -1 & -1
\end{array}
\tag{10}
$$

is 1, which leads to a total of two dimensionless groups, one governed and one governing. The same result is obtained adopting a system with the only dimension of flux, say $\Phi$, for which the dimension matrix

$$
\begin{array}{c|ccc}
 & \mathrm{ET} & R & \mathrm{ET}_{\max} \\
\hline
\Phi & 1 & 1 & 1
\end{array}
\tag{11}
$$

also has rank 1.

If one chooses $R$ as repeating variable, then applying the Pi theorem gives the so-called Budyko's hypothesis

$$\Pi_{\mathrm{B}} = \varphi_{\mathrm{B}}(D_I), \tag{12}$$

where

$$\Pi_{\mathrm{B}} = \frac{\mathrm{ET}}{R} \quad \text{and} \quad D_I = \frac{\mathrm{ET}_{\max}}{R} \tag{13}$$

is Budyko's dryness (or aridity) index. If instead one chooses $\mathrm{ET}_{\max}$, then the Turc's hypothesis follows (Daly et al., 2019)

$$\Pi_{\mathrm{T}} = \varphi_{\mathrm{T}}(H_I), \tag{14}$$

where

$$\Pi_{\mathrm{T}} = \frac{\mathrm{ET}}{\mathrm{ET}_{\max}} \quad \text{and} \quad H_I = \frac{R}{\mathrm{ET}_{\max}} = \frac{1}{D_I}. \tag{15}$$

is the humidity index. Which representation to use is mostly a matter of convenience, since they are related by $\Pi_{\mathrm{T}} = \Pi_{\mathrm{B}}/D_I$ and $\varphi_{\mathrm{T}} = \varphi_{\mathrm{B}}/D_I$. However, Budyko's hypothesis may be more suitable to emphasize the dry end of the hydrologic spectrum, while Turc's privileges the humid end.





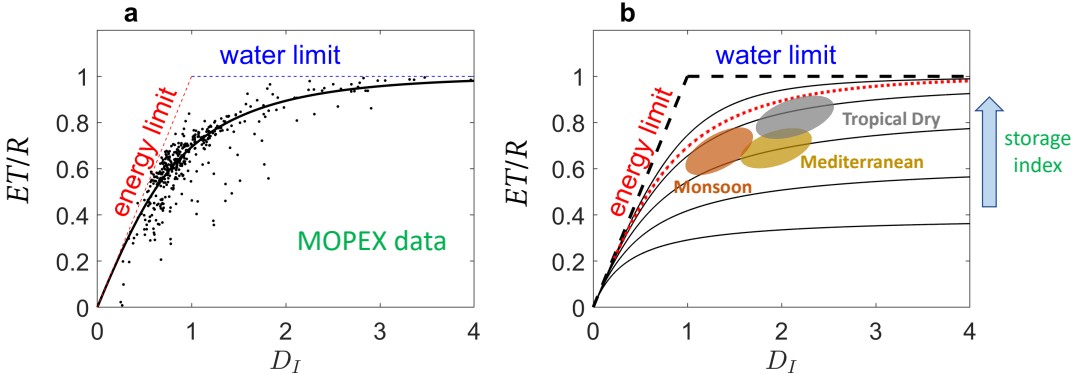

**Figure 2.** Rainfall partitioning. a) Test of Budyko's hypothesis using the MOPEX data (Porporato and Yin, 2022). The solid line shows the original semi-empirical curve used by Budyko, $\Pi_{\mathrm{B}} = (D_I \tanh D_I^{-1}(1 - e^{-D_I}))^{1/2}$. b) Different solutions of the mean partitioning of the minimalist soil moisture balance model of Porporato et al. (2004), showing the role of the storage index, $\gamma = w_0/\alpha$, and seasonality (Feng et al., 2012). The red dotted line is the original curve of Budyko.

The Budyko's hypothesis is tested in Figure 2 using the MOPEX data for several river basins in the continental USA, and
plotted along with the original Budyko's interpolating curve. Clearly, as shown many times before, the main controls on the rainfall partitioning are captured by the dryness index.

### 3.1.2 Storage index: the role of the hydrologic active depth and the variability timescale

While the dryness index captures the main variability of the data, the scatter in Figure 2 also suggest additional controls, which prompt us to revisit the physical law (9) by considering additional governing quantities. There are several quantities that capture
the different properties of the hydrologic system and its climatic forcing, such as storage capacity, areal extension, vegetation type, seasonality, etc.. Among these candidates, the storage depth (the storage volume divided by the area) is probably the first one to consider, as it appears when considering the dramatic effects on the rainfall partitioning when paving a vegetated field.

To account for this effect, we may add a quantity $w_0$ with dimensions of length (L), that is a volume divided by an area. This leads to a physical law of the type

$$\mathrm{ET} = f_{\mathrm{ET}}(R, \mathrm{ET}_{\max}, w_0). \tag{16}$$

However, when trying to apply the Pi theorem, it becomes immediately apparent that, if $w_0$ is chosen as a repeating variable, it actually drops from the dimensionless formulation, similarly to the way the mass disappears from the list of variables when applying the Pi theorem to the pendulum (Barenblatt, 1996). Physically, this tells us that to see a difference in the partitioning among the hydrologic balance of a parking lot, a crusted soil, and a deep vegetated soil, the role of the soil depth must be
associated to other variables, which include a time scale to account for the variability of the hydrologic balance, for otherwise all the other influences are already contained in the dryness index!





Thus, to achieve a further refinement in the Pi-theorem formulation, in combination with the storage depth $w_0$ Porporato et al. (2004) introduced a timescale related to the mean time between rainfall events, or –better– its inverse, the frequency of rainfall events, say $\lambda$. Since the frequency of rainfall times the mean rainfall depth per event $\alpha$ is equal to the mean rainfall rate 280 $R = \alpha\lambda$, the presence of the mean rainfall rate in the list becomes redundant if one already includes the mean rainfall depth $\alpha$. With these additions, the physical law becomes

$$\mathrm{ET} = f_{\mathrm{ET}}(\lambda, \alpha, \mathrm{ET}_{\max}, w_0), \tag{17}$$

with the dimension matrix of rank 2 given by

|   | ET | $\lambda$ | $\alpha$ | $\mathrm{ET}_{\max}$ | $w_0$ |
|---|---|---|---|---|---|
| L | 1 | 0 | 1 | 1 | 1 |
| T | $-1$ | $-1$ | 0 | $-1$ | 0 |

$$\tag{18}$$

It is instructive to explore the full range of choices related to the three possible dimensionless groups, given by the five variables minus the rank two of the matrix. In turn, this implies five possibilities (of the six potential pairs of repeating variables, $(\alpha, w_0)$ obviously must be excluded because of dimensional dependence), which can all be brought back to the dryness index $D_I$ and the storage index $\gamma = w_0/\alpha$, as shown in the following table:

| Repeating var. | Pi theorem | Group relation |
|---|---|---|
| $(\alpha, \lambda)$ | $\frac{\mathrm{ET}}{\alpha\lambda} = \varphi_1\left(\frac{E_{\max}}{\alpha\lambda}, \frac{w_0}{\alpha}\right)$ | $\Pi_\mathrm{B} = \varphi_1(D_I, \gamma)$ |
| $(\lambda, E_{\max})$ | $\frac{\mathrm{ET}}{E_{\max}} = \varphi_2\left(\frac{\alpha\lambda}{E_{\max}}, \frac{w_0\lambda}{E_{\max}}\right)$ | $\Pi_\mathrm{B} D_I^{-1} = \varphi_2(D_I^{-1}, \gamma D_I^{-1})$ |
| $(\alpha, E_{\max})$ | $\frac{\mathrm{ET}}{E_{\max}} = \varphi_3\left(\frac{E_{\max}}{\alpha\lambda}, \frac{w_0}{\alpha}\right)$ | $\Pi_B D_I^{-1} = \varphi_3(D_I^{-1}, \gamma)$ |
| $(\lambda, w_0)$ | $\frac{\mathrm{ET}}{\lambda w_0} = \varphi_4\left(\frac{E_{\max}}{\lambda w_0}, \frac{\alpha}{w_0}\right)$ | $\Pi_\mathrm{B} \gamma^{-1} = \varphi_4(D_I \gamma^{-1}, \gamma^{-1})$ |
| $(E_{\max}, w_0)$ | $\frac{\mathrm{ET}}{E_{\max}} = \varphi_5\left(\frac{\alpha}{w_0}, \frac{\lambda w_0}{E_{\max}}\right)$ | $\Pi_\mathrm{B} D_I^{-1} = \varphi_5(\gamma^{-1}, \gamma D_I^{-1})$ |

$$\tag{19}$$


These five hydrologic spaces provide different, if related, scaling laws that allow us to focus on the role of specific combinations of parameters and emphasize different hydrologic conditions. It is useful to note that the dimensionless group $\vartheta = \gamma D_I^{-1} = \frac{\lambda w_0}{E_{\mathrm{ET}max}}$ appears often as an independent variable. The latter can also be written as $\vartheta = \frac{\lambda}{\eta}$, which uncovers an interpretation of it as a ratio of timescales, one related to the mean time between rainfall occurrence, $1/\lambda$, and one related to the time of 295 depletion of the soil store reservoir of depth $w_0$ at the maximum evapotranspiration rate, i.e. $\eta = E_{\max}/w_0$. Note also that $\vartheta = \frac{\lambda}{\eta}$ appears naturally in the normalized form of the evolution equation of the probability distributions of soil moisture dynamics in minimalist stochastic models (Porporato et al., 2004; Rodríguez-Iturbe and Porporato, 2007). Figure 2b reports different partitioning curves for different values of the storage index $\gamma$, obtained by solving a minimalist stochastic soil moisture model with only four dimensional parameters as in Eq. (17).

With the goal of analyzing the suitability of different hydrologic spaces to capture the information available in global datasets on hydrologic partitioning, Daly et al. (2019) analyzed a physical law focused on hydrologic fluxes (i.e., rates) only. Accordingly, they considered an extension of Budyko and Turc physical law (9) by hypothesizing an additional control by a governing





variable $\phi$ having the dimensions of a flux, $[\phi] = \mathrm{LT}^{-1} = \Phi$,

$$\mathrm{ET} = f_{\mathrm{ET}}(R, \mathrm{ET}_{\max}, \phi). \tag{20}$$

Since each of the fluxes (L/T=$\Phi$) can be used as a repeating variable, three hydrologic spaces are obtained. Assuming further that $\phi$ is a flux related to the storage capacity and the frequency of rainfall, $\phi = \lambda w_0$, the three spaces (Daly et al., 2019) are found to coincide with specific cases in (19): case 1 if $R$ is chosen as repeating variable, case 2 if $\mathrm{ET}_{\max}$ is chosen, and case 4 when $\phi$ is chosen.

The analysis by Daly et al. (2019) shows that by accounting for the ability of a catchment to store water to supply evapotran-
spiration, the flux $\phi$ (referred to as maximum storage rate) serves as a modulator for the relationships of ET with $R$ and $\mathrm{ET}_{\max}$ for very dry and very wet catchments. In this way, accounting for it, it allows us to expand the Budyko and Turc frameworks, suggesting that they are not equivalent, as often assumed in parametric models, unless $\phi \to \infty$. Thus the variable $\phi$ helps group catchments with similar evapotranspiration rates with respect to different combinations of the dimensionless groups $\phi/\mathrm{ET}_{\max} = (\lambda w_0)/\mathrm{ET}_{\max} = \vartheta = \gamma D_I^{-1}$ and $\phi/R = w_0/\lambda = \gamma$, which account for key catchment and hydrologic character-
istics. It also facilitates the analysis of the partitioning in catchments with intermediate values of the dryness index, while Budyko's and Turc's hypotheses help in the analysis of very dry and wet catchments, respectively (Daly et al., 2019).

### 3.1.3 Seasonality, variable coevolution, and higher order effects

It is logical to wonder about the effects of adding other potentially important variables to the physical law of rainfall parti-
tioning. These could include variables describing seasonality, soil and vegetation properties, and other details of the climatic
forcing. With the goal of investigating the role of seasonality in rainfall and evapotranspiration, Feng et al. (2012) considered the duration of the wet season and the intensity of rainfall seasonality. The effect of the new dimensionless groups obtained on the long-term partitioning is shown in Figure 2b. Depending on the degree and type of seasonality, a reduction of evapo-transpiration is typically observed, due to an increase in percolation during the wet season compared to the homogeneous case with no seasonality. At the level of approximation afforded by the long-term averaging, the effects of seasonality are hardly
distinguishable from those of a decrease in the storage index $\gamma$. To disentangle the various effects, and more clearly see the effects of seasonality, the temporal variability in the rainfall partitioning must be considered. An example of this is presented in Figure 3, where the Budyko curve is plotted parametrically as a function of time and the non uniqueness related to seasonal storage becomes evident (Feng et al., 2015).

It is clear that more detailed analyses to disentangle the role of different ecohydrological variables on the rainfall partitioning
should focus on specific aspects of the space-time variability of evapotranspiration, which instead are lost in the spatially lumped, long-term evapotranspiration rates of Budyko's type analyses. The combination with simple physically based models may be a valuable way to sharpen the hypotheses of the physical laws, especially when trying to unravel the effects of the covariation of some of the variables. These regard questions of how the rooting depth, and thus the storage capacity of the active soil layer, may depend on the dryness index and whether such covariations may imply some adaptation of vegetation and
soil properties to the hydroclimatic characteristics. Along these lines, the minimalist stochastic model developed by Porporato



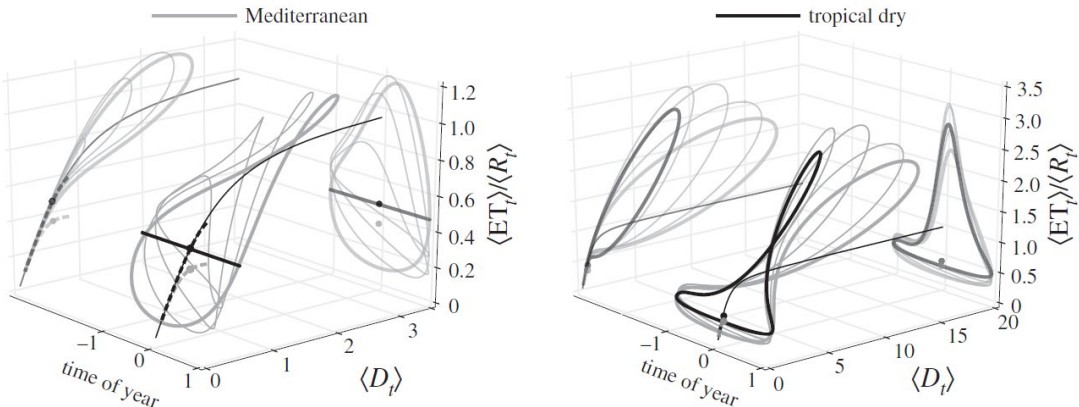

**Figure 3.** Transient trajectories of the time varying ratio $\langle \mathrm{ET}_t \rangle / \langle R_t \rangle$ as function of the time varying dryness index $\langle D_t \rangle$ and time of year for Mediterranean and tropical try climates. Brackets indicate ensemble averages for a given time of year, taken over the rainfall variability due to a time dependent (i.e. seasonal) marked Poisson process of rainfall and seasonal potential evapotranspiration. Each loop is derived from the same climate inputs with the exception of the phase difference between rainfall and potential evapotranspiration, with the thick grey lines showing results for out-of-phase and thick black lines for in-phase. The +1 on the time of year axis corresponds to when maximum rainfall occurs (wet season) and -1 corresponds to timing of minimum rainfall (dry season). The annual average value for each loop falls on a single point on the annual Budyko's curves, shown as dashed lines (black and grey) for those that account for climate seasonality, and as thin black lines for the classical curve which considers only annually averaged climate values. After Feng et al. (2015).

et al. (2004) pointed to a tendency, for the points lying on the Budyko's curve, to converge towards typical values of the storage index of 5-6, in turn suggesting a root-depth adaptation to the intensity of rainfall. Similarly, Or and Lehmann (2019) attributed the convergence of the rainfall partitioning to typical evaporative depths, while Hunt (2021) connected a dependence of the storage index with hydroclimatic characteristics, $\gamma(D_I)$, to an adaptation brought about by constraints linked to percolation

statistics.

Shedding light on the role of higher order controls on rainfall partitioning requires charting more detailed model-data investigations, branching off the beaten path of Budyko's type analyses. Long-term averages may only contain a weak signal of those interactions, which may be easily overwhelmed by noise and other data limitations. This suggests a need for adapting dimensional analysis to these other aspect of the rainfall partitioning and formulating more sophisticated physical laws that

capture the more subtle controls by the soil-plant-atmosphere system.

## 3.2 Soil formation, weathering, and loss

A partitioning of the soil mineral component also takes place, on much larger time scales, on the land surface (Riebe et al., 2004). The loss of minarls at the land surface is controlled by series of tightly connected chemico-physical processes with interesting geomorphology and hydrologic interactions (Maher and Chamberlain, 2014), which in turn have a crucial impact





on the surface energy and water balances, as well as several biogeochemical, ecological and climate processes (Garrels, 1983; Richter and Billings, 2015).

The input of solid material to the soil from breaking up parent rock is called rock denudation. While the dissolved minerals and the very fine particles resulting from denudation can be transported away by water, the remaining particles stay in the ground where they continue to be weathered, until they too can be transported away. As the soil ages and transforms chemically,

the soil may also be deformed by the action of internal stresses and move as creeping flow. As a result, if one assumes, for simplicity, that the input and output balance (Riebe et al., 2004), denudation $D$ equals soil formation rate. These in turn balance the losses by weathering $W$ and erosion plus creep, which are lumped here into a single term $E$ (Figure 1), yielding the following soil partitioning,

$$D = W + E. \tag{21}$$

To analyze this partitioning and help resolve the complex interaction between chemical weathering, climate, and the hydrologic cycle, Calabrese and Porporato (2020) used dimensional analysis. This alloed them to obtain a theoretical framework to organize the existing data on weathering rates. Because of the crucial role of leaching of dissolved inorganic carbon to the sites of weathering, the problem strongly depends on the rainfall partitioning at the surface, discussed in the previous section. Accordingly, focusing on the weathering rate as the governed quantity, Calabrese and Porporato (2020) wrote a physical law

assuming that weathering rates are function of the input given by the denudation rate, a maximum weathering rate rate (which includes the effects of type of parent material, temperature, etc.), the concentration of dissolved inorganic carbon [DIC], and the percolation flux LQ (see Eq. (3)). Further assuming that both [DIC] and LQ only depend on the surface rainfall partitioning through the dryness index, as in the Bydyko's hypothesis (12), they wrote

$$W = f_W(D, W_{\max}, D_I). \tag{22}$$

Formally, apart form the presence of the dimensionless dryness index, the situation is similar to the one of Eq. (3), since $W$, $D$, and $W_{\max}$ have all the same dimensions of a flux of mineral per time. Choosing the input $D$ as the repeating variable in Eq. (21), as Budyko did in the rainfall partitioning, the Pi theorem gives

$$\frac{W}{D} = \varphi_W\left(\frac{W_{\max}}{D}, D_I\right). \tag{23}$$

To verify this functional relationship, Calabrese and Porporato (2020) used literature data including granitic, basaltic, and

shale terrains and encompassing a broad range of environmental settings. After normalizing the data by their maximum rate and assuming for the range of data available that $W \leq D$, the results (Figure 4a) show a remarkable linear trend of the type

$$\frac{W}{D} = \left(\frac{W_{\max}}{D}\right)^m \psi_W(D_I). \tag{24}$$

Since the exponent $m$ of the power law is very close to 1, the expression can be further simplified to the final form

$$\frac{W}{W_{\max}} = \psi_W(D_I). \tag{25}$$





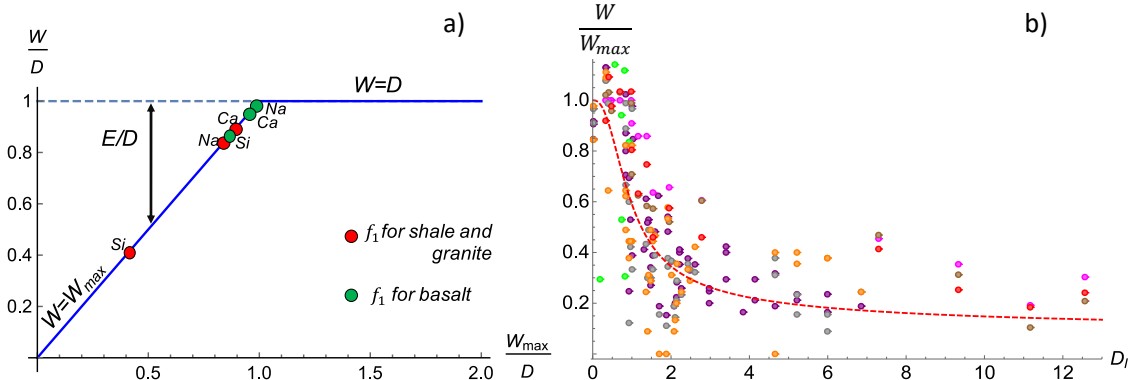

**Figure 4.** a) Analysis of the self-similarity hypothesis in Eq. (24); b) Specific weathering rates as a function of the dryness index; dashed line shows Eq. (26). After Calabrese and Porporato (2020).

Both these expressions are suggestive of self-similarity. At a first glance, looking at (24), one would be led to conclude that we are in the presence of self-similarity of the second type with respect to $W_{\max}/D$. However, an exponent practically equal to one, unlike the typical irrational numbers expected in incomplete self similarity (Barenblatt, 1996), leads to a simplification which eliminates the governing variable $D$. As a result, the final expression (25) can be interpreted as a case of complete self-similarity with respect to $D/W_{\max}$, when $W_{\max}$ is chosen as a repeating variable in (22) instead of $D$. Alternatively, the same

expression can be obtained directly from the Pi theorem starting from an abbreviated physical law $W = f_{W_a}(W_{\max}, D_I)$. The precise meaning of this self-similarity remains an open problem to be investigated. The physical message however is that the denudation rate is not a relevant quantity compared to the maximum weathering.

     Equation (25) provides a mathematical structure to analyze how the water cycle affects the chemical depletion fraction through the dryness index. The empirical data fit provides an equation with only one parameter

$$\psi_W(D_I) = 1 - \frac{\ln(D_I^\alpha + 1)}{1 + \ln(D_I^\alpha + 1)}, \tag{26}$$

which emphasizes the strongly nonlinear relation between water availability and weathering rate (see Figure 4b). Starting from high $D_I$ values, the normalized weathering rate increases only slightly up until $D_I \sim 2$, after which it steeply increases before plateauing at $D_I \sim 0.5$. The increase region corresponds to the establishment of grassland, savanna, and shrubland ecosystems, emphasizing the important role of vegetation in acidifying the soil and in turn enhancing weathering rates. A similar nonlinear

dependence on wetness has been observed in the global distribution of soil organic carbon (Kramer and Chadwick, 2018).

     The scatter showed by the data in Figure 4b is not unexpected, because of the large datasets used, which are obtained from different locations and different methods. While it is likely that part of it may be attributable to noise, it would also be interesting to consider whether adding other quantities in the physical law might help explain more of this variability. A logical starting point could be the analysis of the effects of the soil depth and water-storage capacity, which could be explored by correlating

the data with estimates of the storage index $\gamma$. It is likely that this will reveal further linkages between surface hydrology, soil formation, and weathering.





### 3.3 Landscape self-similarity

Besides the macroscopic effects of wetness on weathering discussed in the previous Section, the loss of minerals driven by the coupled water and sediment fluxes on the topographic surface is also responsible for the formation of complex topographic

patterns, which in turn impact ecohydrologic processes (Dietrich and Perron, 2006). In this Section, we focus on the dendritic morphology of interlocked ridges and valleys, the self-similarity of which is linked to the fractal behavior of landscapes (Rodríguez-Iturbe and Rinaldo, 2001). While on the one hand the complexity of these patterns hinders analytical results, on the other hand it also helps with the investigation of macroscopic (i.e., averaged) behavior, giving rise, as we will see, to emergent scaling behaviors for the large scale statistics of the sediment budget, the mean elevation profile, and the landscape spectral

properties.

Towards the goal of performing dimensional and self-similarity analyses, here we will consider only (spatially) averaged quantities in idealized geometries, for which the solutions may be expected to depend only on a limited number of dimensionless quantities. The effort of formulating meaningful physical laws for average landscape quantities is facilitated by the existence of simplified, semi-empirical landscape evolution models (Chen et al., 2014), which can be inspected to infer the main

governing quantities of the problems. We specifically refer to a minimalist landscape evolution model in detachment limited conditions (Howard, 1994), although many considerations apply also to transport limited and other intermediate formulations (Davy and Lague, 2009; Pelletier, 2012). Accordingly, the equation for the evolution of the landscape elevation $z(x,y,t)$ is (Chen et al., 2014)

$$\frac{\partial z}{\partial t} = \delta \nabla^2 z - K a^m |\nabla z| + U, \tag{27}$$

where $t$ is time, $\delta$ is the diffusion coefficient used to represent the intensity of soil creep, $K$ is the fluvial erosion coefficient, $a(x,y,t)$ is the specific drainage area, $m$ is a dimensionless coefficients (for simplicity we assume the exponent of the gradient in the erosion term to be equal to 1), and $U$ is the surface-growth term due to tectonic uplift. Eq. (27) is coupled to the 'conservation' equation for the specific drainage area

$$\nabla \cdot \left( a \frac{\nabla z}{|\nabla z|} \right) = 1. \tag{28}$$

The latter was derived by Bonetti et al. (2018) as an idealized representation of the water flow, following steepest descent lines of topographic surface with given characteristic speed (see also Bonetti et al., 2020). Mathematically, $a = \lim_{w \to 0} A/w$, where $A$ is the contributing area and $w$ is a finite portion of a contour (iso-elevation); hence, the specific drainage area has dimensions of length, $[a] = \text{L}$, and is defined at a point on the landscape, whereas the contributing area $A$, $[A] = \text{L}^2$, is actually zero when considered at a point, unless the topographic surface is discontinuous (see Bonetti et al., 2018, for details). The use of $A$

instead of $a$ in (27) is therefore incorrect and leads to grid-dependent results in mathematical codes. The coupled equations (27) and (28) form a closed system once the initial and boundary conditions are specified. Figure 5 shows simulation results producing complex patterns (Anand et al., 2020) that resemble real landscapes, with characteristic branching and channelization instability.



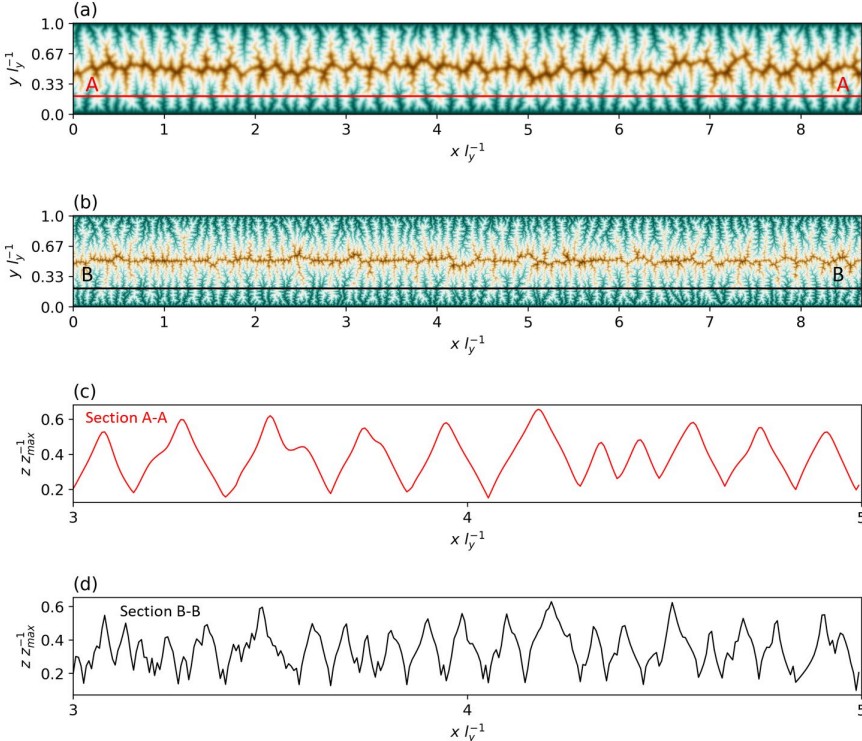

**Figure 5.** Steady state simulation of landscape elevation given by (27) and (28) for $m = 0.5$ and different channelization indices, Eq. (33), $\mathcal{C}_I = 10^3$ in (a) and $\mathcal{C}_I = 10^4$ in (b). The elevation along A-A and B-B transects is shown in (c) and (d). After (Hooshyar et al., 2021).

### 3.3.1 Sediment budget and soil partitioning

In steady state, spatial averaging leads to the partitioning of soil material into soil creep (SC) and the soil loss by stream erosion (SL),

$$U = \mathrm{CR} + \mathrm{SL}. \tag{29}$$

This equation is obviously related to Eq. (21), since at steady state the rate of uplift needs to equal the denudation rate, while the losses may be grouped by focusing on different features of the underlying processes. Eq. (21) emphasizes chemical weathering,

while lumping all the other losses (mainly erosion and creep) into the term $E$, while here Eq. (29) draws attention to the water-erosion term SL (possibly including also weathering), while the rest is lumped into a diffusive term, CR, representing soil creep and other smoothing processes.





Averaging spatially (27) and (28), the mean sediment balance equation can be written as (see Hooshyar and Porporato, 2021, for details)

$$
\quad \underbrace{\delta(2l_y)^{-1}\langle\nabla z_\perp\rangle_\Omega}_{-\mathrm{CR}} - \underbrace{K\left((1-m)\langle\kappa_c a^m z\rangle + m\langle a^{m-1}z\rangle\right)}_{\mathrm{SL}} + U = 0, \tag{30}
$$

where $l_y$ is the domain width, $\nabla z_\perp$ is the component of the elevation gradient normal to the domain contour $\Omega$, $\kappa_c$ is the plane or contour curvature, the brackets $\langle\cdot\rangle$ indicate spatial average, and $\langle\cdot\rangle_\Omega$ is the average over the domain contour. While this is an exact result, the terms in brackets are unknown because of the complexity of the landscape surface. This is where dimensional analysis comes in handy.

Bonetti et al. (2020) tackled the problem of these spatial averages by adopting a directional dimensional analysis (see Sec. 2.3), considering independent dimensions for horizontal lengths, L, and the vertical length, $\mathrm{L}_z$. Physically, a justification for the use of directional dimensional analysis may be found in the fact that Eq. (27) is the sediment budget equation at a point written in terms of volume of sediments per unit ground area. For sediments of constant density, the same equation can be written in term of mass per unit ground area, say $\rho z$, which then can be considered as a new variable with its independent dimensions. Indicating the unknown average slope at the boundary as $S_* = \langle\nabla z_\perp\rangle_\Omega$, the physical law can be written as

$$
S_* = f(l_y, K, U; \delta, m), \tag{31}
$$

where the various quantities are suggested by inspection of the governing equations and the type of boundary conditions. For this problem, $[S_*] = \mathrm{L}_z\mathrm{L}^{-1}$, $[U] = \mathrm{L}_z\mathrm{T}^{-1}$, $[\delta] = \mathrm{L}^2\mathrm{T}^{-1}$, $[K] = \mathrm{L}^{1-m}\mathrm{T}^{-1}$, $[m] = 1$. Choosing $l_y$, $K$, and $U$ as repeating variables, the Pi theorem then yields

$$
\quad \frac{\delta S_*}{U l_y} = \frac{2\mathrm{CR}}{U} = \varphi_{\mathrm{CR}}(C_I^{-1}, m) = \psi_{\mathrm{CR}}(C_I, m), \tag{32}
$$

where

$$
C_I = \frac{K l^{m+1}}{\delta} \tag{33}
$$

is the global channelization index. Figure 6 shows the plot of the function (32) obtained from a set of simulations with different $C_I$ and $m$ values.

These results suggest a tantalizing analogy with the analysis of wall-bounded turbulent shear flows. In fact, the regular behavior of $\varphi(C_I, m)$ reminds of the behavior of the Darcy friction factor plotted in the Moody diagram for pipe flow (Munson et al., 2013). For a detailed description of dimensional analysis in turbulence, see Barenblatt (1996) and Katul et al. (2019). In this analogy, the slope of the elevation profile and the slope of velocity profile play a similar role. As well known, in turbulence, the mean velocity profile at the wall is proportional to the wall shear stress and relates to the partitioning of viscous and turbulent stresses; this is analogous here to the partitioning of soil coming from uplift into creep and stream erosion – see Eq. (30). We will highlight further links with the dimensional analysis of wall-bounded turbulent flows in the subsequent developments.

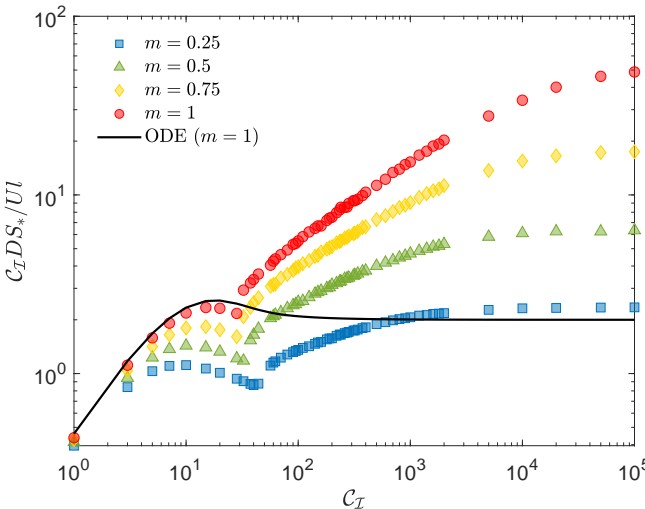

**Figure 6.** Average sediment partitioning according to Eq. (32) as a function of the channelization index for different values of $m$. Points refer to numerical simulations and solid line refers to the analytical solution for the unchannelized regime (courtesy of Sara Bonetti).

It is also useful to note that in Bonetti et al. (2020) the Pi theorem was applied to (31) considering $l_y$, $U$, and $\delta$ as repeating variables, which instead leads to $\Pi_\delta = \varphi_\delta(C_I)$, with $\Pi_\delta = \Pi_{\mathrm{CR}} C_I$ and $\varphi_\delta(C_I) = \psi_{\mathrm{CR}}(C_I) C_I^{-1}$. In the turbulence analogy,

the present analysis corresponds to choosing viscosity instead of density (which is the typical choice; see Katul et al. (2019)) as one of the repeating variables in the physical law for the wall-shear stress. Thus, for the wall-sear stress, $\tau = \mu \Sigma_*$, where $\mu$ is the dynamic viscosity and $\Sigma_*$ is the slope of the streamwise velocity profile at the wall, the physical law is $\tau = f_\tau(V, L, \rho, \mu, \epsilon)$, where $\rho$ is the density, $V$ the mean velocity, $L$ the characteristic lateral dimension, and $\epsilon$ the roughness height. Using $V, L, \mu$ as repeating variables, one has $\frac{\tau L}{\mu V} = \mathrm{Re} \cdot \frac{\tau}{\rho V^2} = \mathrm{Re} \cdot f_{\mathrm{Darcy}}\left(\mathrm{Re}, \frac{\epsilon}{L}\right)$, where $\mathrm{Re} = \frac{\rho V L}{\mu}$ is the bulk Reynolds number.

A second interesting fact is the plateauing of the curves at large values of $C_I$, for both the analytical solution in the unchannelized case and the numerical results in the channelized regime. This suggests the existence of complete self-similarity for $C_I \to \infty$, again analogous to the self-similarity observed in the Moody diagram for the friction factor in the fully rough regime (Munson et al., 2013), implying that while erosion dominates over creep, the effect of diffusion does not fully disappear but remains present, probably concentrated in an area of zero measure corresponding to the network of ridges and valleys. Again, this

singular limit bears similarities with the much investigated limit of the viscous Navier-Stokes equations for Reynolds numbers tending to infinity as well as the hypothesis of dissipative anomalies in the inviscid Euler equations (e.g., Eyink, 2008).

### 3.3.2   Mean elevation profile

Hooshyar et al. (2020) analyzed the profile of the mean elevation, $\bar{z}(y) = \lim_{l_x \to \infty} \frac{1}{l_x} \int_0^{l_x} z(x, y)\, dx$, of channelized landscapes. Based on the inspection of the governing equations and using again directional dimensional analysis, the physical law





for the slope of the profile was assumed as

$$\frac{d\bar{z}}{dy} = f_{\frac{d\bar{z}}{dy}}\left(y, \delta, z_*; l_y, K, U, m\right), \tag{34}$$

where $y$ is the distance from the boundary, $z_*$ is an elevation scale. Choosing $y$, $\delta$, and $z_*$ as repeating variables and with simple manipulation of the dimensionless groups, the Pi theorem gives

$$(m+1)\eta\frac{d\varphi}{d\eta} = f_3\left(\eta, \mathcal{C}_\mathcal{I}, \zeta, m\right), \tag{35}$$

where $\varphi = \frac{\bar{z}}{z_*}$. In addition to the global channelization index, $C_I$, reflecting the relative impact of fluvial erosion to diffusive transport, $\eta = \frac{Ky^{m+1}}{\delta}$ is a local variable with a similar form as $\mathcal{C}_\mathcal{I}$ but capturing the local relative contribution of those two processes, while $\zeta = \frac{Ul_y^2}{\delta z_*}$ describes the relative impact of tectonic uplift to diffusive transport.

In a system with relatively small diffusive transport and dominated by erosion and uplift, $\mathcal{C}_\mathcal{I}$ and $\zeta$ take high values. The same argument also applies to $\eta$ except for locations at an intermediate distance from the boundary. Thus, when the variables $\eta$,

$\mathcal{C}_\mathcal{I}$, and $\zeta$ reach such an asymptotic condition, one may assume complete self-similarity (Barenblatt, 1996) according to which the function $f_3$ is independent of these quantities

$$\eta\frac{d\varphi}{d\eta} = \kappa\left(m\right), \tag{36}$$

where $\kappa$ is only a function of $m$. Integrating Eq. (36) yields

$$\varphi = \kappa\left(m\right)\ln\eta + C, \tag{37}$$

where $C$ is independent of $\eta$ but may still depend on $m$, $\mathcal{C}_\mathcal{I}$, and $\zeta$. Eq. (37) describes the logarithmic scaling of the mean-elevation profile with respect to $\eta$. The emergence of such a logarithmic profile was confirmed in numerical simulations, laboratory experiments, and real landscapes, as well as in other models of complex branching such as the optimal channel networks and directed percolation (Hooshyar et al., 2020). Figure 7 shows the flattening of mean elevation profiles with increasing $C_I$ (recall that a similar effect is observed also in turbulent velocity profiles), along with the values of the coefficient $\kappa$ of the

logarithmic profile.

As observed by Hooshyar et al. (2020), the presence of a mean logarithmic profile of elevation at an intermediate distance from the domain boundaries is similar to the classic results for the turbulent velocity profile (see also Barenblatt, 1996). In wall-bounded turbulence (Katul et al., 2019), the logarithmic profile for the mean velocity profile is obtained from the Pi theorem applied to the physical law for the mean gradient, $\frac{d\bar{u}}{dy} = f(y, u_*, \rho, L, \mu, \epsilon)$, where $u_* = \sqrt{\frac{\tau}{\rho}}$ is the friction velocity (the other

quantities have the same meaning as in the previous Subsection) and then assuming complete self-similarity with respect to both the bulk and local Reynolds numbers. From a phenomenological point of view, the analogy between the two phenomena is found in the resemblance between progressive penetration to smaller scales of ridges and valleys into the landscape and the intensification of vorticity producing smaller and smaller vortices (i.e., smaller Kolmogorov scales). The increased turbulent mixing and the progressive land-surface dissection with sharper sequences of ridges and valleys surface dissection with in-

creasing channelization index and Reynolds number, respectively, produces in both cases a flattening of mean profile observed





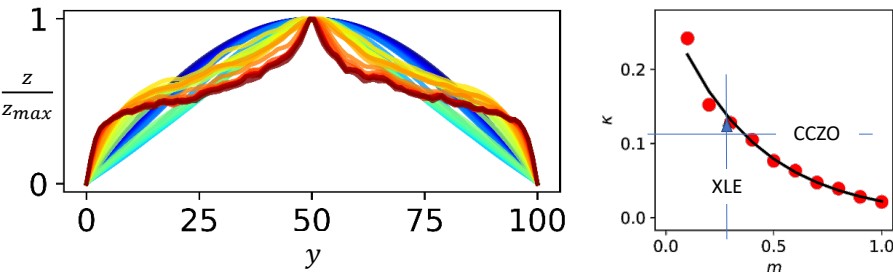

**Figure 7.** Mean elevation profiles for increasing values of $C_I$ (from blue to red lines) and, on the left, coefficient of the logarithmic profile as a function of the exponent $m$ for simulations, the XLE laboratory experiments (Hooshyar et al., 2019), and the landscape at the Calhoun Critical Zone Observatory (CCZO). Modified after (Hooshyar et al., 2020).

in the turbulence mean velocity and the mean elevation (see Fig. 7a) as well as a logarithmic scaling. The fact that a logarithmic region is found in different contexts, including directed percolation (Hooshyar et al., 2020), hints at the generality of such self-similar scaling as a robust outcome in dynamically different complex systems, appearing as a dimensional consequence of length-scale independence in spatially bounded complex systems.

### 3.3.3 Spectral analysis of elevation fluctuations

Our final example follows Hooshyar et al. (2021) who analyzed the power spectral density (PSD) of elevation transects at given $y$ (see Figure 5),

$$\mathcal{P}_y(\omega) = \frac{1}{l_x}|\hat{z}_y(\omega)|^2, \tag{38}$$

for $l_x \to \infty$, where

$$\hat{z}_y(\omega) = \int_0^{l_x} z_y(x)e^{-2\pi i\omega}dx \tag{39}$$

is the Fourier transform of $z_y(x)$ and $\omega$ is the longitudinal wavenumber or inverse scale. These PSDs peak at a wavenumber (i.e., the most energetic mode) which provides a characterization of the typical valley spacing. Beyond such wavenumbers, a power-law scaling is typically visible in simulations, producing an asymptotic behavior which can be collapsed onto a single curve at high $\mathcal{C}_I$.

Hooshyar et al. (2021) analyzed the PSD spectral scaling with the aid of directional dimensional analysis. They argued that for a longitudinal elevation series at a given $y$ the amount of 'energy' (i.e., variance of the elevation fluctuations) at a wavenumber $\omega$ must depend on the following variables

$$\mathcal{P}_y(\omega) = g_1\left(\omega, y, l_y, \delta, K, U, m\right), \tag{40}$$



where $g_1$ is the physical law. The energy $\mathcal{P}_y(\omega)$ is defined over the fluctuations along the $x$-axis (see Eq. (38)) and has dimension $\mathrm{L}_z^2\mathrm{L}$. The wavenumber along the $x$ direction has dimension $[\omega] = \mathrm{L}^{-1}$, while $\delta$, $K$, and $U$ have dimensions $\mathrm{L}^2\mathrm{T}^{-1}$, $\mathrm{L}^{1-m}\mathrm{T}^{-1}$, and $\mathrm{L}_z\mathrm{T}^{-1}$, respectively; $y$ and $l_y$ have dimension L.

Given three dimensions $\mathrm{L}_z$, L, T and 7 dimensional governing variables, and choosing $K$, $U$, and $\omega$ as repeating variables, the Pi theorem yields

$$\frac{\mathcal{P}_y(\omega)K^2\omega^{3-2m}}{U^2} = g_2\left(\frac{K\omega^{-(m+1)}}{\delta}, \omega l_y, \omega y, m\right). \tag{41}$$

A manipulation of Eq. (41) leads to

$$\frac{\mathcal{P}_y(\omega)K^2\omega^{3-2m}}{U^2} = g_3\left(\mathcal{C}_{\mathcal{I}}, \eta, \eta_\omega, m\right). \tag{42}$$

The quantity $\eta = Ky^{m+1}/\delta$ has the same form as that of $\mathcal{C}_{\mathcal{I}}$ but defined locally at $y$ distance from the boundary, and $\eta_\omega = K\omega^{-(m+1)}/\delta$ is equivalent to $\eta$ but defined in the frequency domain.

In the asymptotic limit of relatively high $\mathcal{C}_{\mathcal{I}}$, $\eta$ and $\eta_\omega$ attains a near-constant limit away from the boundary, implying complete self-similarity,

$$\mathcal{P}_y(\omega) \propto \omega^{2m-3}, \tag{43}$$

where the proportionality coefficient is $(U/K)^2 g_3(m)$. Eq. (43) predicts the exponent of the power spectral density that is independent of $\delta$ and has a power-law decay. The condition of high $\mathcal{C}_{\mathcal{I}}$, $\eta$ and $\eta_\omega$ is expected in systems that are dominated by erosion (high $\mathcal{C}_{\mathcal{I}}$), far enough from the boundary (high $\eta$), and within small enough scales (high $\eta_\omega$). Again a parallel with the Kolmogorov spectrum of turbulence (Pope, 2001; Katul et al., 2019) can be drawn with the channelization index playing the role of the Reynolds number. Moreover, at a sufficient distance from the domain boundary, it may be assumed that the information regarding the domain geometry and direction is lost, becoming statistically isotropic. This is similar to the local isotropy at small scales (or eddies detached from the boundary) of fully developed turbulent flow.

Fig. 8a shows the exponent of the power fits to PSDs for simulations with $\mathcal{C}_{\mathcal{I}} \geq 10^5$, denoted by $\alpha$, for different $m$ values. This finding agrees with the relation $\alpha = 2m - 3$ in the intermediate range of $m$ and supports the validity of the assumption of complete self-similarity with respect to $\eta_\omega$. The inset of Fig. 8b also shows the function $g_3(m)$ from numerical simulations. The spectral scaling was confirmed also in the laboratory experiments and the real topography at the Calhoun Critical Zone Observatory (Fig. 8c-f).

From a geomorphological point of view, the connection between the exponent $\alpha$ and the parameter $m$ in the erosion term provides a useful link to landscape processes, since steep landscapes with debris-flow-dominated channels have been associated to smaller $m$, while flatter fluvial landscapes are characterized by larger values of $m$ (Montgomery and Foufoula-Georgiou, 1993). The values of the spectral exponent is also interesting from a more theoretical point of view, in relation to the role of nonlinearities as a function of scale. Our analysis has shown consistently values of $\alpha \sim -2.5$, while several studies previously reported exponents near $-2$ (Newman and Turcotte, 1990; Huang and Turcotte, 1989; Passalacqua et al., 2006); the latter corresponds to a fractal dimension $D_m = 1.5$ (Huang and Turcotte, 1989; Voss, 1985) and to the presence of an underlying





fractional Brownian noise (Turcotte, 1987; Bell Jr, 1975) with a Lorentzian spectrum and an exponential decay of the elevation autocorrelation. These models with $\alpha = -2$ would therefore imply linear stochastic dynamics, at odds with the known presence of nonlinear terms responsible for the very formation of the channel network.

On the one hand, $\alpha = -2.5$ would mean an erosion exponent $m = 0.25$, therefore preserving the nonlinearity of the dynamics

also for mascroscopic scaling relationships, like the PSD scaling. On the other hand, the fact that the observed value is not far from $\alpha = -2$, might mean that averages taken over complex landscape patterns cause an effective reduction of nonlinearity, whereby the activation of many degrees of freedom at high channelization regimes causes a statistical regularity which muffles the small scale nonlinearities. This is certainly an interesting topic that deserves further investigation.

Finally, from a practical point of view, the spectral scaling of landscape elevation could be profitably utilized in developing

efficient numerical simulations of landscape evolution (Passalacqua et al., 2006). Such numerical schemes would potentially resemble large eddy simulation methods used in fluid turbulence Pope (2001), where the unsolved dynamics at finer scales are approximated by extrapolating the PSD. The improved speed would be a great asset for large scale simulations of landscape evolution under different scenarios.

## 4  Conclusions

We have presented several examples of applications of the Pi theorem and self-similar scaling to the partitioning of water and sediments driven by the terrestrial water cycle. It is time to draw to a close and ask ourselves whether by using dimensional analysis we have learned anything useful regarding these problems. An answer in the affirmative is suggested by considering that dimensional analysis helped reveal the dominant controls of the dryness index and storage index in the long term rainfall partitioning, while in the weathering analysis it allowed us to extract the key nonlinear control of dryness index, making order in

the vast amount of information contained in global datasets from different experiments and environmental conditions. Finally, the analysis of the complex geometries obtained from landscape evolution models allowed us to discover new macroscopic relations among macroscopic variables in the partitioning of sediments, the elevation profile, and the spectral scaling. The analogy between landscape elevation and turbulent velocity fluctuations has also been fruitful and promises further results.

If these observations confirm the utility of dimensional analysis, it should also be clear that these methods are not a fool-

proof set of rules to achieve miraculous solutions. Rather, they are an iterative procedure to sharpen our hypotheses on physical processes. Thinking of dimensional arguments as a form of modelling allows an 'explication of the role abstraction and multiple realisability; not as compatibility with other possible worlds but as compatibility with different fictional descriptions of our own world' (Pexton, 2014). We hope that these considerations will help breathe new life in Strahler's view (Strahler, 1958) that dimensional analysis will become increasingly useful in ecohydrology, geomorphology, and beyond.

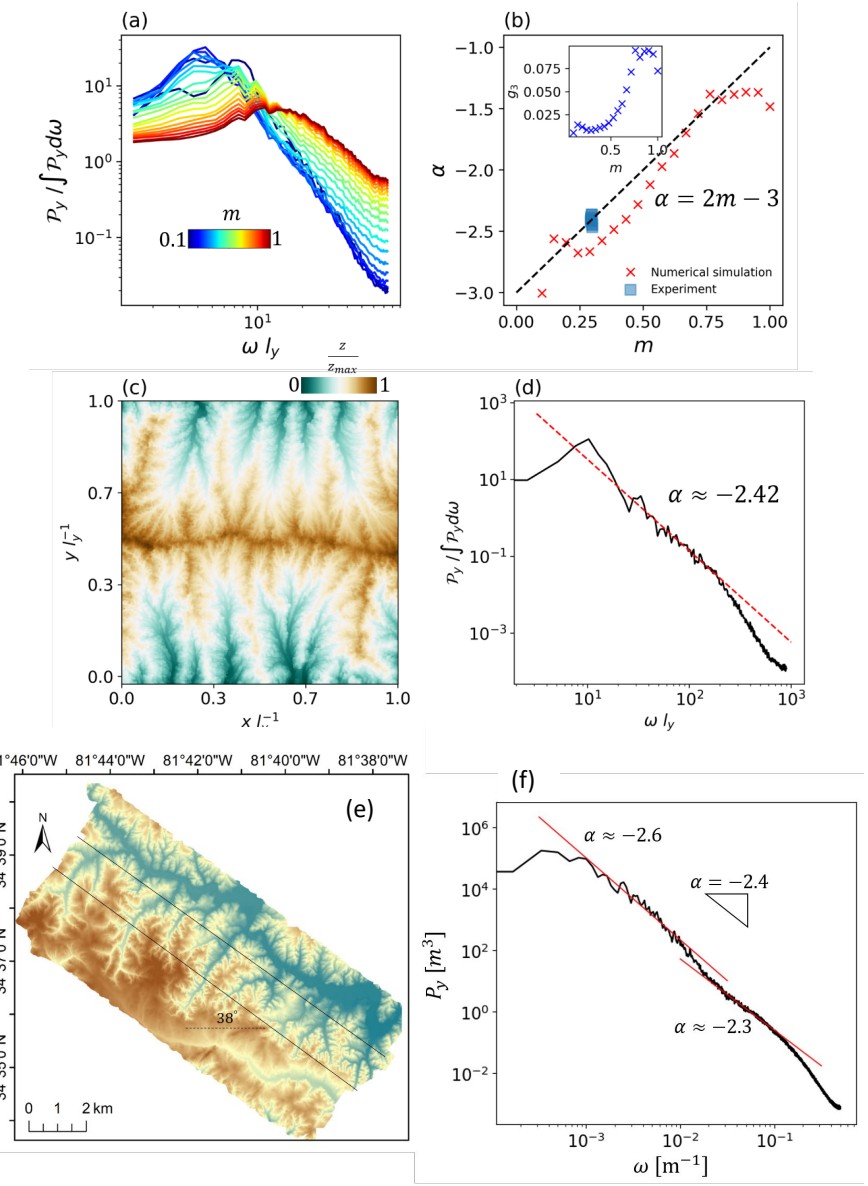

**Figure 8.** (a) PSDs of elevation longitudinal series for different values of $m$ at an intermediate distance from the boundary. (b) Slope of power fit to the declining part of the PSD from simulations with $\mathcal{C}_{\mathcal{I}} \geq 10^5$, denoted by $\alpha$, as a function of the exponent $m$. The data from the XLE physical experiment are also shown. The black line is the relation $\alpha = 2m - 3$ derived from dimensional and self-similarity arguments in Eq. (43). The inset shows the function $g_3(m)$ from numerical simulations. (c) shows an example of the XLE experimental landscape (Hooshyar et al., 2019), of which (d) shows the PSD. (e) Portion of the landscape at the Calhoun Critical Zone in South Carolina, USA from 1-m resolution LiDAR data. (f) The PSD is computed from 1-D transects within the area contained by the two parallel lines that are located at an intermediate distance from the main channel shows two distinct power-law scaling regions with similar exponent. After Hooshyar et al. (2021).





## Appendix A: The Rayleigh-Riabouchinsky controversy: thermodynamic limit as complete self-similarity

In this appendix we discuss in more detail the Rayleigh-Riabouchinsky Controversy related to augmented dimensional analysis. In particular, after a brief historic background, we show how the thermodynamic approach of Rayleigh corresponds to a self-similar solution of the first kind with respect to the dimensionless group obtained form the Boltzmann constant, which serves as the dimensional unifier when augmenting the dimensional analysis from mechanics to thermodynamics.

As mentioned in the main text, Rayleigh (1915a)[1] considered the problem the heat flux $h$, the governed quantity, between a body of characteristic dimension $a$ and a stream of incompressible and inviscid fluid moving with velocity $v$. Besides $a$ and $v$ the other governing quantities are the specific heat capacity $c$ and the thermal conductivity $\kappa$. Rayleigh formulated a physical law with $n = 5$ governing variables and adopted length, time, energy, and temperature as primary dimensions (i.e., $k_{\mathrm{Ray}} = 4$), as typical in thermodynamics (of course one could use also LMT plus temperature, using mass instead of energy). As a result, he obtained one ($n - k_{\mathrm{Ray}} = 5 - 4 = 1$) governing dimensionless group, which he used to successfully describe the problem.

In a very short comment published soon after Rayleigh's paper, Riabouchinsky (1915) objected that adopting the more advanced and detailed point of view of statistical mechanics, according to which temperature is the mean molecular kinetic energy, one could more parsimoniously limit the primary dimensions to length, time and energy, without involving temperature. As a result, he obtained two ($n - k_{\mathrm{Ria}} = 5 - 3 = 2$) dimensionless groups. In his response, Rayleigh rejected Riabouchinsky's alternative commenting that: 'It would indeed be a paradox if the further knowledge of the nature of heat afforded by molecular theory put us in a worse position than before in dealing with a particular problem. The solution would seem to be that the Fourier equations embody something as to the nature of heat and temperature which is ignored in the alternative argument' (Rayleigh, 1915b).

Rayleigh's reply was authoritative and sensible, but not completely satisfactory. As pointed out by Buckingham (1915): 'since he does not pursue the subject further and the reader may feel as if left in mid-air, it seems worth while that the point raised by M. Riabouchinsky should be somewhat further elucidated.' Since then the debate has been subject of continued discussions. Here, taking a cue from previous analyses of this controversy (e.g., Gibbings, 1980; Butterfield, 2001) and adopting the use of a dimensional unifier (Panton, 2006), we show that Rayleigh's result may be seen as a self-similar solution of the first kind of the augmented physical law obtained by including the Boltzmann constant as a dimensional unifier to link thermodynamics to mechanics. Starting from a thermodynamic approach to the problem is tantamount to taking for granted the existence of this limit.

Thus, following Rayleigh, we specifically choose length (L), time (T), energy (heat, H), and temperature ($\Theta$) as primary dimensions. With $[h] = \mathrm{HT}^{-1}$, $[a] = \mathrm{L}$, $[\theta] = \Theta$, $[c] = \mathrm{HL}^{-3}\Theta^{-1}$, $[\kappa] = \mathrm{HL}^{-1}\mathrm{T}^{-1}\Theta^{-1}$, $[v] = \mathrm{LT}^{-1}$, and $[k_B] = \mathrm{H}\Theta^{-1}$, we

---

[1]This Nature paper contains the famous quote of Lord Rayleigh: "I have often been impressed by the scanty attention paid even by original workers in physics to the great principle of similitude. It happens not infrequently that results in the form of 'laws' are put forward as novelties on the basis of elaborate experiments, which might have been predicted a priori after a few minutes' consideration."


obtain the dimension matrix

$$
\begin{array}{c|ccccccc}
 & h & a & \theta & c & \kappa & v & k_B \\
\hline
L & 0 & 1 & 0 & -3 & -1 & 1 & 0 \\
T & -1 & 0 & 0 & 0 & -1 & -1 & 0 \\
H & 1 & 0 & 0 & 1 & 1 & 0 & 1 \\
\Theta & 0 & 0 & 1 & 0 & -1 & 0 & -1
\end{array}
\tag{A1}
$$

of rank 4. As a result, the Pi theorem gives 7-4=3 dimensionless groups.

Choosing the heat flux $h$ as the governed quantity and selecting $a$, $\theta$, $c$, and $\kappa$ as dimensionally independent (repeating) variables among the governing quantities leads to the physical law

$$
h = f(a, \theta, c, \kappa; v, k_B),
\tag{A2}
$$

from which the Pi theorem then gives

$$
\frac{h}{d\theta\kappa} = \varphi\left( \frac{vcd}{\kappa}, \; k_B d^3 c \right).
\tag{A3}
$$

With a system of units for which $k_B = 1$ one obtains the result of Riabouchinsky (1915),

$$
\frac{h}{d\theta\kappa} = \varphi_{\mathrm{Ria}}\left( \frac{vcd}{\kappa}, \; d^3 c \right).
\tag{A4}
$$

This however would imply the use of very small units, because in the usual SI system, $k_B = 1.380649 \times 10^{-23}$ J $\cdot$ K$^{-1}$. As a result, apart from systems at the nanoscale, in normal conditions $k_B d^3 c \sim 0$. Our everyday experience, on which thermodynamic concepts are based, shows that we can neglect this term in (A3) to arrive at the result, originally obtained by Rayleigh,

$$
\frac{h}{d\theta\kappa} = \varphi_{\mathrm{Ray}}\left( \frac{vcd}{\kappa} \right).
\tag{A5}
$$

Thus, looking at Eq. (7), it becomes clear that the thermodynamic limit, which allows us to go from statistical mechanics to continuum mechanics and thermodynamics, corresponds to a complete self-similar solution of (A3) with respect to $k_B d^3 c$. Alternatively, the same result is obtained by straightforward application of the Pi theorem to the restricted physical law

$$
h = f_{\mathrm{Ray}}(a, \theta, c, \kappa; v),
\tag{A6}
$$

as originally done by Rayleigh.

*Competing interests.* The author declares no competing interests.

*Acknowledgements.* This article follows a kind invitation by EGU related to the Dalton medal presentation by the author. Much of this work has been the result of invaluable collaborations with Edoardo Daly, Jun Yin, Xue Feng, Salvatore Calabrese, Sara Bonetti, Milad Hooshyar,





and Shashank Anand. The author is also grateful to Paolo D'Odorico, Gaby Katul, Luca Ridolfi, and Ignacio Rodriguez-Iturbe for continued friendship, support, and advice. The US National Science Foundation (NSF) grant nos. EAR-1331846 and EAR-1338694, the BP through the Carbon Mitigation Initiative (CMI) at Princeton University, and the Moore Foundation are acknowledged for financial support.



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
