# Peer review of "Hydrology without Dimensions\"

_Hydrology and Earth System Sciences, 2021_

## Referee Comment (RC2)

General Assessment: As the title already suggests, this manuscript considers the inference of hydrological laws based on a combination of physics, boundary conditions, and self-similarity arguments (complete and incomplete) using the Buckingham Pi theorem.  The illustrations selected focus on processes with rich complexity that simply prevent the use of first principles.  In this sense, they offer a strong test case for the use (and usefulness) of the Buckingham Pi theorem.  To be clear, the work here is precisely in the spirit of what the Buckingham Pi theorem intends to achieve – a compact representation of a wealth of data (i.e. dimension reduction) in the form of a similarity law.  The illustrations here include 3 examples – the Budyko curve (multiple version depending on the complexity of rainfall representation), weathering rates and mineral partitioning, and the spectral properties of landscapes (that is an insightful analysis of a PDE representing processes necessitating closure schemes that can be derived from the Buckingham Pi).  The work concludes with analogies between landscape and turbulent spectral cascades and the equivalent role of the Reynolds number in elevation profiles (i.e. channelization index) thereby opening up new vistas to the study of landscape evolution.

The introduction part is well crafted and elegantly summarizes a number of topics based on complete and incomplete similarity arguments, as well as an insightful explanation of directional dimensional analysis.  The paper concludes with conjectures about how the Buckingham Pi theorem be used in an era where data generation is overwhelming the ability of data interpretation.  It is safe to state that this new perspective differs from the text-book illustration of the Buckingham Pi in hydrology and hydraulics – which is the drag coefficient as a function of Reynolds numbers for flow over a sphere or the Darcy-Weisbach friction factor as a function of bulk Reynolds number and relative roughness in pipes.  The diversity of sources and references is quite complete – and covers depth and breadth for students and researchers alike.  For this reason, I am recommending that the manuscript be accepted with technical comments and friendly suggestions.

Minor comments:

1. The introduction: the dichotomy between the promise of dimensional analysis and the number of results in the geosciences derived from the Pi theorem is certainly illuminating.  But, perhaps it is also wise to point out that a large number of problems in the geosciences – when tackled in their most general form, lead to a large number of Pi groups.  Simply put, the usage of the Pi theorem becomes tenuous when n-k>>1.  In these cases, a restricted version of the problem must be sought that may enable the Buckingham Pi to be applied as an effective dimension reduction approach.  The cases selected here for illustration are clearly chosen so that n-k is a small number (order unity).  Even an experienced chef may have hard time predicting the taste of a dish when the number of ingredients is large and the number of steps involved in a recipe is no less large!. In those situations, focusing on certain statistics or long time scales or restricted range of length scales (or modes of activity) must be sought.  Whether these length or time scales are the pertinent ones remains problem specific.  This is where the 'art' of using the Buckingham Pi comes into play.  This point was alluded to in lines 230-238 as well as lines 329-335, and lines 341-345, but warrants a general statement early on.

2. Example 3.3 takes on a disproportionate space in the paper and serves as an illustration when the continuous space-time dynamics are known through an approximated PDE (eq. 27). I wonder whether this section can benefit from some re-arrangements. To arrive at the PDE, a number of assumptions were invoked that themselves could have benefited from the Buckingham Pi (e.g. the erosion term). Hence, the Buckingham Pi can be used as a 'closure' scheme to construct the governing PDE of elevation. Next, certain aspects of this resulting PDE can then be analyzed using the Buckingham Pi. This would be similar to what Prandtl did when invoking K-theory to close the turbulent stress term in the Reynolds-averaged Navier-Stokes equation thereby arriving at an advection-diffusion equation for turbulent boundary layer flows. This averaging and closure scheme enabled Prandtl to convert the elliptic nature of the instantaneous (and unsolvable) Navier-Stokes equation to a parabolic PDE (when time averaged) that can be solved. In essence, the erosion term is way too complicated to derive from first principle but the closure used here resulted in an approximated but closed PDE (i.e. eq. 27) with coefficients that are difficult to derive solely from first principles (much like eddy diffusion) that require experiments. The remaining sections become an analysis of solutions after some averaging and homogenizations are applied.

For example, section 3.3.1 proceeds to deals with simplifications leading to bulk expressions whereas section 3.3.3 considers the scaling laws and spectral properties of the stationary state in a restricted region.

This example can be a prototypical case study of how an approximated PDE can be constructed to describe the space-time evolution of the system when adopting certain closure schemes (derivable from Buckingham Pi). Then, proceed to show how one may proceed to derive different physical laws depending on the a priori assumptions made about the space and time scales and regions of the solution. The point here is that the tactic used to arrive at the approximated PDE and the method of analysis of this PDE can serve as a methodological complement to the elegant justification given in Figure 1.

---

## Referee Comment (RC3)

**Review report on "Hydrology without dimensions"**

by Demetris Koutsoyiannis

| | |
|---|---|
| Author(s): | Amilcare Porporato |
| Manuscript title: | Hydrology without dimensions |
| Journal: | Hydrol. Earth Syst. Sci. Discuss. |
| Journal's Ref.: | hess-2021-442 |
| Reviewer's Ref.: | DK-JR-374 |
| Date: | 2021-09-27 |
| Recommendation: | Accept as is |

**Reviewer's assertion**: It is my opinion that a shift from anonymous to eponymous (signed) reviewing would help the scientific community to be more cooperative, democratic, equitable, ethical, productive and responsible. Therefore, it is my choice, consistent with my aesthetic attitude, to sign my reviews. Furthermore, I believe that the current trend in the review system to seek credit for anonymous transactions (by asking recognition for anonymous reviews through Publons) is problematic on ethical and aesthetic grounds.

**Reviewer's clarification**: The references included in this review have the same meaning that references have in scientific documents. In brief, they clarify or justify the reviewer's statements and provide links where further details can be found. They are not necessarily meant to be suggestions for the author(s) to include them in the paper in review (if not already included).

As stated in the acknowledgments, this article by Amilcare Porporato is a contribution invited by EGU and related to the Dalton medal lecture by the author.[1] The depth and breadth of the analyses and the presentation clearly reflect the fact that Porporato is a pioneering scientist well deserving the Dalton medal.

I have no reservation to recommend this article for publication in its current form. On the other hand, for the sake of the dialogue, I tried to find some points of disagreement and focus my discussion on them. I am offering my comments just for the author's consideration and not necessarily as suggestions for changes in the paper.

Overall, the choice of the paper's theme is very successful. While scientists working in hydraulics and hydrodynamics are familiar with dimensional consistency and dimensional analysis, this is not always the case in hydrology. We often see in hydrological texts empirical equations that are dimensionally inconsistent, dependent on the choice of units or even wrong. Therefore, I believe the paper is didactic and quite useful in this respect. On the other hand, I must acknowledge that it is not an easy read and at times it becomes a difficult one. I understand that this is because it summarizes a lot of knowledge and examines cases from diverse fields.
* * *
[1] I believe this should be stated as a footnote in the first page of the paper.

I particularly liked the section "2.3 Augmented and directional dimensional analysis" and its Appendix examining the question whether temperature can be used as a primary dimension or, alternatively, we should limit the primary dimensions to length, time and energy. However, I had some difficulty to accept the following argument (in Appendix A):

> *This however would imply the use of very small units, because in the usual SI system, $k_B = 1.380649 \times 10^{-23}$ J·K$^{-1}$. As a result, apart from systems at the nanoscale, in normal conditions $k_B d^3 c \sim 0$. Our everyday experience, on which thermodynamic concepts are based, shows that we can neglect this term in (A3) …*

where equation (A3) is:

$$\frac{h}{d\theta\kappa} = \varphi\left(\frac{vcd}{\kappa},\ k_B d^3 c\right)$$

Since we do not know the function $\varphi(\ )$, I think we cannot know the influence of its second argument ($k_B d^3 c$). In my view, whether or not it is very small does not say anything to us. Perhaps, in the unknown function $\varphi(\ )$, this term could be multiplied by a very large constant and give a considerable effect (even if it were an additive term, which we do not even know). I would understand an argument that the term can be omitted because it is virtually constant, or because it is small *and* additive. But as formulated now, I find it problematic. With all this I do not mean that Rayleigh's (1915) result is wrong; I just think that better reasoning is needed.

Perhaps the framework of augmented and directional dimensional analysis could be explained better. An example (perhaps again in the form of an Appendix) additional to the examined temperature question, but closer to hydrology, would be helpful for a reader to understand the framework of augmented dimensional analysis. (I am thinking of a case of a flow or a wave where the vertical direction ($z$) could be regarded independent of the horizontal one ($l$); intuitively this agrees with what we are doing in many of our engineering drawings where we use different horizontal and vertical scales.)

I wonder if the framework of augmented dimensional analysis could also be used as a justification of equation (7), instead of introducing a power law (self-similarity) out of the blue. Personally, I had been amazed by scaling behaviours and power laws decades ago, but progressively, I shaped the opinion that scaling claims need proper foundation in order to stand, and there is no magic in their emergence. As I have tried to show (Koutsoyiannis, 2014; Koutsoyiannis et al., 2018), their emergence is understandable as asymptotic laws, whose exponents can hardly coincide in the lower and upper limits. In other words, expressions like equation (7) could only hold asymptotically. If they hold generally, then there must be some theoretical reason that should have to be explained (perhaps in the frame of augmented dimensional analysis?).

Generally, scaling, self-similarity and fractal behaviour look to be overemphasized or overpraised in the paper, perhaps unjustifiably. For example, below Figure 4 it is stated "Both these expressions are suggestive of self-similarity". However, I have some difficulty to locate self-similarity both in the expressions and the figure. I rather see a curve similar

to the Budyko curve in Figure 2 (yet less smooth). Asymptotically this curve seems to have a constant slope on the left and a constant value (zero slope) on the right. Perhaps I have missed the point, but I would not think that such a curve reflects self-similarity. Also, contrary to what is stated in the paper, I do not see "self-similarity […] in the Moody diagram for the friction factor in the fully rough regime".

Since the deterministic relationships examined do not look to be in perfect agreement with the data (see e.g. Figure 2), I would expect some involvement, or at least mentioning of stochastics (by way of replacing one-to-one relationships with many-to-many). But this is my personal taste, which perhaps the author does not share.

I like the fact that the paper cites good old works, starting from the 1880s with Lodge (1888; I guess it would be a real headache for him to combine in his paper the different length units he uses, miles, yards, feet, inches, and products thereof!), and Williams (1890). On the other hand, while Kolmogorov's ideas are mentioned (and named) several times, I think it is a pity that he is not cited at all. Furthermore, Strahler (1958) is perhaps miscited; I doubt if he envisaged that "that dimensional analysis will become increasingly useful in ecohydrology" (did he know the term "ecohydrology"?). The modern literature is well represented, yet I think that the recent papers by Theodoratos et al. (2018), and Theodoratos and Kirchner (2020, 2021) whose subject is dimensional analysis on landscape evolution are relevant and could be cited.

As regards the presentation, the paper is carefully written with very few typing errors which will certainly be spotted in the proof-reading phase. (To mention just one typing error, in the caption of Figure 7, with "on the left", is it not meant "on the right"?) The notation and terminology are both fine. An exception in terminology is perhaps the terms "climatic forcing" which looks not to be meant as such (see e.g. the meaning of this term in Wikipedia). Perhaps just "climate" or "atmospheric processes" are more accurate within the paper's scope.

**References**

Koutsoyiannis, D.: Random musings on stochastics (Lorenz Lecture), AGU 2014 Fall Meeting, San Francisco, USA, doi: 10.13140/RG.2.1.2852.8804, American Geophysical Union, 2014.

Koutsoyiannis, D., Dimitriadis, P., Lombardo, F., and Stevens, S.: From fractals to stochastics: Seeking theoretical consistency in analysis of geophysical data, *Advances in Nonlinear Geosciences*, edited by A.A. Tsonis, 237–278, doi: 10.1007/978-3-319-58895-7_14, Springer, 2018.

Lodge, A.: The multiplication and division of concrete quantities, General Report, Association for the Improvement of Geometrical Teaching, 14, 47–70, 1888.

Rayleigh, L.: The principle of similitude, *Nature*, 95, 66, 1915a.

Strahler, A.N.: Dimensional analysis applied to fluvially eroded landforms, *Geological Society of America Bulletin*, 69, 279–300, 1958

Theodoratos, N. and Kirchner, J.W.: Dimensional analysis of a landscape evolution model with incision threshold, *Earth Surf. Dynam.*, 8, 505–526, doi: 10.5194/esurf-8-505-2020, 2020.

Theodoratos, N. and Kirchner, J.W.: Graphically interpreting how incision thresholds influence topographic and scaling properties of modeled landscapes, *Earth Surf. Dynam. Discuss.* [preprint; accepted], doi: 10.5194/esurf-2020-45, 2021.

Theodoratos, N., Seybold, H., and Kirchner, J.W.: Scaling and similarity of a stream-power incision and linear diffusion landscape evolution model, *Earth Surf. Dynam.,* 6, 779–808, doi: 10.5194/esurf-6-779-2018, 2018.

Williams, W.: On the relation of the dimensions of physical quantities to directions in space, *Proceedings of the Physical Society of London*, 1874-1925, 11, 357, 1890.

---

## Author Response (AR1)

*Prof Nunzio Romano*

*HESS, Editor*

*Dear Nunzio,*

*Thank you and the reviewers for reviewing my paper. I revised the paper (especially Sec. 3 and the related Appendices) following the reviewers' recommendations, which helped improve the paper. The detailed of the response are in the following pages.*

*I remain at your disposal for any further changes. Sincerely,*

*Amilcare*

**Referee 1**

I am very grateful to referee 1 for the nice comments and positive assessment about the paper. In the revised version, I've added a comment regarding the cases in which the dimensions are not easy to define, as in problems interfacing with social and economical components.

**Referee 2**

I thank the referee 2 for his/her nice comments and friendly advice on the paper. I've revised the paper to include these suggestions. In particular, in the introduction and conclusions, I have commented on the fact that the physical law, as a starting point of the dimensional analysis, should be kept as simple as possible exactly to avoid the problem mentioned by the reviewer of ending up with too many dimensionless Pi groups.

Secondly, I've modified and improved the presentation in Sec. 3. This is also in response to Ref. 4, who also had useful suggestions about that Section of the paper. I've now added an appendix (B) on the derivation of the governing equations for the landscape evolution model as well as one (Appendix C) on the analogy with near wall turbulence.

**Referee 3 (Demetris Koutzoyiannis)**

I would like to thank professor Koutzoyiannis for a very nice and useful review.

With regard to his main points, i've now improved the presentation of self-similarity in the discussion of the Rayleigh-Riabouchinsky controrversy and explained that indeed there is nothing a priori in the function \varphi that would lead to the fact that the smallness of the second Pi terms does not count in the overall function. As correctly stated by the reviewer, it is a matter of experience, an empirical fact, that leads to this emergence of themrodynamics neglecting the disorderly degrees of freeedom of molecular motion.

I've also improved the presentation of Sec. 3 and added the reference to the paper by Theodoratos et al. (2018).

I've commented on the fact that the scaling laws obtained with self-similarity assumptions are only asymptotic, as also mentioned in the paper by Koutzoyiannis et al. (2018), which is now cited in the conclusions. The orginal paper by Kolmogorov is now also cited.

Finally, I've corrected the typos that were kindly pointed out and added a footnote to the title, mentioning that this paper is related to the Dalton-medal lecture. Thank you!

**Referee 4 (Stefan Hergarten)**

Many thanks to professor Hergarten for his kind review and positive criticism. I have considered seriously his comments and followed them to improve the paper, especially Sec. 3.

In particular, regarding the units of the scaling laws (1) and (2), I've added a footnote to explain that in mathematics it is common to refer to dimensionless quantities. This is because the arguments of transcendental functions have to be dimensionless anyway (Barenblatt, 1996). Here however, since we are explicitly talking about dimensions, it is important to remember that this implies the presence of dimensional unit factors that make things consistent. Thank you for pointing this out.

Regarding the specific contributing area and the governing equations of LEMs, I've added Appendix B to better explain the derivation and meaning of this variable. The derivation from the surface water equation makes it clear that one needs a specific variable and not a global variable like A (contributing area, which is not defined at a point in a continuous representation. We hope that this is now clearer and that some of the confusion related to this issue is resolved. Thank you.